



# Correlation of core and downhole seismic velocities in high-pressure metamorphic rocks: A case study for the COSC-1 borehole, Sweden

Felix Kästner[1,2], Simona Pierdominici[1], Judith Elger[2], Alba Zappone[3], Jochem Kück[1], and Christian Berndt[2]

[1]Helmholtz Centre Potsdam, GFZ German Research Centre for Geosciences, 14473 Potsdam, Germany
[2]GEOMAR Helmholtz Centre for Ocean Research Kiel, 24148 Kiel, Germany
[3]ETH Zurich, Department of Earth Sciences, 8092 Zurich, Switzerland

*Correspondence to:* Felix Kästner (felix.kaestner@gfz-potsdam.de)

**Abstract.** Deeply rooted thrust zones are key features of tectonic processes and the evolution of mountain belts. Exhumed and deeply-eroded orogens like the Scandinavian Caledonides allow to study such systems from the surface. Previous seismic investigations of the Seve Nappe Complex have shown indications for a strong but discontinuous reflectivity of this thrust zone, which is only poorly understood. The correlation of seismic properties measured on borehole cores with surface seismic data constrains the origin of this reflectivity. In this study, we compare seismic velocities measured on cores to in situ velocities measured in the borehole. The core and downhole velocities deviate by up to 2 km/s. However, velocities of mafic rocks are generally in close agreement. Seismic anisotropy increases from about 5 to 26 % at depth, indicating a transition from gneissic to schistose foliation. We suggest that differences in the core and downhole velocities are most likely the result of microcracks mainly due to depressurization. Thus, seismic velocity can help to identify mafic rocks on different scales whereas the velocity signature of other lithologies is obscured in core-derived velocities. Metamorphic foliation on the other hand has a clear expression in seismic anisotropy. These results will aid in the evaluation of core-derived seismic properties of high-grade metamorphic rocks at the COSC-1 borehole and elsewhere. In particular, they show that core log seismic integration via synthetic seismograms requires wireline logging data in any but mafic lithologies.

## 1 Introduction

Thrust zones in high-pressure metamorphic rocks are important features in mountain belts. In active fault zones they are often accompanied by devastating earthquakes like repeatedly occurring in the Himalayas (e.g., in 2015 [M=7.9] and 2008 [M=7.9]), which are a potential threat to the local population. Their investigation, therefore, is important to improve our understanding of the deeper orogenic processes and tectonic evolution.

However, such structures are seldom directly accessible and difficult to image. An exception are exhumed systems where most parts of the orogen were deeply eroded and exposed to upper crustal levels. The Scandes in western Scandinavia, a remnant of the mid-Paleozoic Caledonian orogeny, represent such a system. Here, available geophysical investigations of an orogen root can be compared to geological and petrophysical observations from the surface. Reflection seismic data provide





another possibility to investigate these thrust zones and reveal structures of a strong and highly diffuse reflectivity like, for example, observed at the highly metamorphic Seve Nappe Complex in Jämtland, western Sweden.

To better understand the origin of these reflections, we can compare them to the physical properties of the related rocks at depths. This involves comparison of seismic velocities from different measurements and scales with both lithological and
structural characteristics of cored rocks. This concept of integration and cross-calibration of data sets across different scales is well established in sedimentary basins of marine and lake environments (e.g., Bloomer and Mayer, 1997; Miller et al., 2013; Riedel et al., 2013; Thu et al., 2002), where it was successfully used to better characterize the subsurface seismic stratigraphy using both core and downhole logging data. But to date, there are only few studies that adapted this concept in hard-rock environments and metamorphic complexes (e.g., Emmermann et al., 1990; Xu et al., 2009).

The project COSC (Collisional Orogeny in the Scandinavian Caledonides) is a scientific drilling project co-funded by the International Continental Scientific Drilling Program (ICDP), the Swedish Research Council, and the Geological Survey of Sweden. It aims to study the mountain building processes of the Scandinavian Caledonides in western Sweden (Gee et al., 2010; Lorenz et al., 2015a). In 2014, the COSC-1 borehole was drilled in Åre-Jämtland (Fig. 1) to a total depth of 2495.8 m. It was fully cored below 103 m with almost 100 % core recovery. Drilling was accompanied by extensive field campaigns
providing physical properties from downhole logs and borehole and surface seismic profiles (Hedin et al., 2014, 2016; Krauß et al., 2015; Lorenz et al., 2015a; Simon et al., 2015). Together with the excellent core recovery and data availability, the COSC project constitutes a perfect case study to apply core-log-seismic data integration in a metamorphic environment.

The objective of this study is to characterize the impact of structural and compositional variations on seismic properties. We determine seismic properties on core scale (mm to cm) from the COSC-1 borehole in western Sweden and evaluate their
potential to explain the in situ seismic properties (mm to km scale) by comparing core velocities with downhole logging data (cm to m scale) and core lithology. Thereby, we focus on the effects of different ambient pressure conditions on the velocities as well as scale differences inherent in the individual data sets and measurement procedures (Fig. 2).

As core velocities measured under atmospheric pressure conditions can exhibit strong deviations from in situ conditions (Elbra et al., 2011), we integrate and compare the core and downhole seismic velocities with laboratory velocity and anisotropy
results from 16 core samples measured under different confining pressure simulating in situ conditions. Ultimately, this results in better constrained seismic properties in the vicinity of the COSC-1 borehole, which is a prerequisite for a successful core-log-seismic data integration (Worthington, 1994).

## 1.1 Geological Background

In the mid-Paleozoic, the Caledonian orogen formed during the continent-continent collision of Laurentia and Baltica (Gee
et al., 2008). The Scandinavian Caledonides are composed of nappes that were thrusted over the Baltic platform margin accommodating several hundred kilometres of southeast-ward shortening. These thrust sheets are subdivided into the Lower, Middle, Upper, and Uppermost Allochthons (Gee et al., 1985). While the Lower Allochthon is mainly composed of sedimentary rocks of Upper Proterozoic to Silurian age, the Middle Allochthon consists of crystalline metamorphic rocks. The





upper part of the Middle Allochthon comprises the Seve Nappe, a complex with high grade metamorphism of locally early

Silurian eclogite (e.g., Ladenberger et al., 2014). The Upper Allochthon consists of units of greenschist facies dominated by

sedimentary rocks while the Uppermost Allochthon is mostly composed of metasediments and carbonates.

The dimension of the Caledonian mountain range and its formation mechanisms are similar to those of the more recently

formed Himalayan orogen (Labrousse et al., 2010). Subsequent glaciation, tectonic uplift, and gravitational collapse left most

parts of the mountain range deeply eroded exposing rock formations of middle to lower crust levels. Today's remnants, the

Scandes, extend over a distance of about 300 km across the Scandinavian Peninsula, over a length of about 1700 km, from the

Norwegian Skagerrak coast in the South up to the North Cape. An extensive review of the Caledonian Orogeny and related

areas is provided by Corfu et al. (2014) and Gee and Sturt (1985).

Based on first lithological descriptions of the cored rocks, four main sections were identified (Lorenz et al., 2015a): (1)

gneisses of varying compositions (mainly felsic, amphibolitic, calc-silicate), often garnet- and diopside-bearing, occur from

top to about 1800 m; (2) an extensive deformation zone prevails between 1800 m and 2345 m; followed by (3) a 15 m-thick

retrograde transition zone from amphibolite facies gneisses into lower-grade meta-sedimentary rocks; and (4) mylonitized

quartzites and metasandstones of unclear tectonostratigraphic position that characterize the lowermost part to the bottom of

the borehole at 2495 m.

The potential to investigate the deep structure of the orogenic root from the surface is directly addressed in the COSC

drilling project. It focuses at the physical properties and inner structure of the emplaced nappe complex associated with high-

grade metamorphic allochthonous rocks as well as the character and age of deformation of the underlying thrust sheets, the

main Caledonian décollement and the Precambrian basement (e.g., Lorenz et al., 2015a).

## 2    Data and Methods

For this study, we used compressional wave velocities from a multi-sensor core log (MSCL) and downhole logging data

from the COSC-1 borehole. In addition, we measured selected core samples to provide seismic properties of characteristic

lithological units. Based on these laboratory measurements, we calculated velocities at different environmental conditions

(intrinsic, lithostatic pressure, and atmospheric pressure), which then served as a calibration tool for the core and downhole

logging data. The individual data sets and experimental acquisition used for this study are described in the following

subsections. Table 1 provides an overview of the individual measurements and related velocities, which nomenclature is used

throughout this study.

### 2.1    Laboratory measurements

Laboratory analyses are routinely applied to study the elastic properties, fabric, and seismic anisotropy of crustal and mantle

rocks (e.g., Barberini et al., 2007; Kern, 1982; Siegesmund et al., 1991; Zappone et al., 2000). We used seismic wave velocities

measured on 16 core samples. These measurements were conducted at room temperature and varying confining pressure using



the pulse-transmission method (Birch, 1960, 1961). In order to determine the seismic anisotropy, i.e., the directional
       dependence of seismic velocity, compressional (P) wave, the velocities were measured on three mutually perpendicular core
       plugs drilled out of each of the 16 core samples.

       Six of these samples were measured by Wenning et al. (2016), chosen based on the most abundant lithologies derived from
       the lithological core description (Lorenz et al., 2015b). In order to extend and complement these measurements, we selected
ten additional samples at depth intervals and lithologies that were not previously covered (see Table A1 for a detailed sample
       list). We chose samples from zones of higher and lower reflectivity indicated by the zero-offset vertical seismic profile (Simon
       et al., 2015).

       The core samples were cut at lengths of about 15 to 20 cm from the COSC-1 drill cores. From each sample, we drilled
       three cylindrical core plugs of 3 to 5 cm length and 2.54 cm diameter. The orientation of the plug axes $x$, $y$, and $z$ agree with
the major structural axes that are defined by the sample's foliation and lineation, and which we have determined by visual
       inspection. Following the common practice (Zappone et al., 2000), we designated $z$ as the axis normal to the foliation plane.
       The foliation plane is spanned by the $x$- and $y$-axes, where $x$ is oriented parallel and $y$ perpendicular to the apparent lineation.
       The core plug ends were cut and ground to plane-parallel surfaces in order to provide a good coupling with the signal
       transducers. The plugs were oven-dried at 100 °C for at least 24 h in order to eliminate free water from the pore space.

The plug dimensions (length and diameter) and dry weight were determined using a micrometer caliper (+/- 0.01 mm) and
       precision balance (+/- 1e-3 g), respectively. We used the average diameter and length of four successive readings to reduce
       possible errors due to surface irregularities. The matrix density of each core plug was measured using a He-gas pycnometer
       (type: AccuPyc II 1340). This is based on a precise volume measurement using the gas displacement method (Lowell et al.,
       2004, p. 326).

The experimental procedure was similar to the one described in Wenning et al. (2016). For each core plug, ultra-sonic
       seismic velocities were acquired under different confining pressure using a hydrostatic pressure vessel and corresponding
       acquisition system (Fig. 3; Barblan, 1990). The setup consisted of two piezoelectric transducers (lead-zircon ceramics, 1 MHz
       resonance frequency), transmitter and receiver, which were placed on the core plug's cylinder faces. They were hold in place
       by a shrink tube jacketing plug and transducers. Additional metal wires, tightly wrapped around the shrink tube, sealed the
core plug to prevent any oil leakage into it (Fig. 3). The prepared core plug was mounted inside the vessel where the oil pressure
       was applied and controlled using a pneumatic compressor. The pressure was first increased in 50 MPa steps, from 50 to 250
       MPa, and then decreased in 30 MPa steps, between 240 and 30 MPa (±2 MPa). At each pressure step, a wave generator
       connected to one of the transducers produced an input square signal 0.2 microseconds wide, with an amplitude of 30 Volts,
       and a pulse rate of 0.5 kHz. Simultaneously, the wave generator sent a trigger signal with the same frequency to a PC-based
wave analyzer. The analyzer used an impedance of 1 MΩ over a range of ±500 mV. The waveforms were recorded with a
       repetition rate of 80 ns and a sampling rate of 100 MHz. The electric noise was minimized by averaging the tracks.

       The measurements were calibrated using steel cylinders of varying lengths to correct for the delay in the observed travel
       times (i.e., $t_{observed} = t_{rock} + t_{system}$), which was caused by the cables, transducers, and interfaces in the electronic system. The



calibration was conducted at confining pressures of 50 and 100 MPa. The system travel time ($t_{system}$) was obtained by averaging

the results from these two pressures.

We performed our measurements at room temperature (ca. 22 °C), which should be a good approximation of the in situ condition because of the very low geothermal gradient (ca. 20 °C/km) and low temperatures observed at about 2500 m, the bottom of the borehole ($T_{log}$ < 60 °C; Lorenz et al., 2015b).

In the subsequent data processing, we calculated velocities and anisotropy coefficients as a function of confining pressure

for each core plug and sample, respectively. Because of a very good signal quality, it was not necessary to apply any additional filtering to the waveform data. First-arrival times were picked manually using a picking tool developed for this purpose (Grab et al., 2015). The seismic velocities were calculated using the plug length $L$, divided by the corrected travel time: $v = L / (t_{picked} - t_{system})$. Changes in the plug length due to compression can be neglected for these rock types (Zappone et al., 2000).

We calculated P-wave velocities at atmospheric and at lithostatic pressure to relate to the different conditions of the velocity

measurements from downhole sonic, multi-sensor core logger (MSCL), and borehole seismic data. The lithostatic pressure was calculated from the core and downhole logging density and is shown together for the associated sample depths in Fig. 4. Since density was only logged down to about 1600 m, we used extrapolated densities down to 2500 m depth showing slightly higher pressures than those calculated from the core.

We used the relationship derived by Ji et al. (2007) to calculate velocity-pressure curves for each core plug (Fig. 5). This

relationship consisted of a four-parameter exponential equation to relate the measured velocities to confining pressure by solving a least-squares curve-fitting problem. The intrinsic velocity (Vp0), which corresponds to the undisturbed, crack-free rock matrix, was calculated from the intercept of the extrapolated linear part of the velocity-pressure relation. The velocities at room pressure (VpAP, p = 0.1 MPa) and lithostatic pressure (VpLP, p = Sv) were determined from the non-linear representation of the velocity-pressure relation (Fig. 5). The six samples from Wenning et al. (2016) applied a slightly different

relation proposed by Wepfer and Christensen (1991). For most applications such as the linear part and the intrinsic anisotropy this relation provides very similar results. However, based on a rather empirical relationship and purely mathematical curve-fitting approach, its ability to fit the fully-measured velocity-pressure curves is lower than the physically-derived model (Ji et al., 2007). Moreover, the zero boundary condition made it unfavourable for extrapolations of the measured velocities to zero confining pressure.

The intrinsic seismic anisotropy (AVp) was determined from the Vp0 measured along the x, y and z directions. In literature there are different representations of the velocity anisotropy (Birch, 1961; Crampin, 1989; Schön, 1996). We used a definition after Crampin (1989), which is described by the degree of the fractional difference of the maximum and minimum velocity of the rock sample, i.e., $AVp = 100*(Vp_{max} - Vp_{min})/Vp_{max}$.

The main source of error in the determination of seismic velocity is the uncertainty in picking the first arrival times. We

estimated a typical uncertainty in the picking of the P-wave first arrivals with an upper limit of $\delta t_{observed} \cong \pm 0.1$ μs. The seismic velocity uncertainty was estimated using the concept of error propagation (e.g., Taylor, 1997), providing an measured P-wave velocity uncertainty of $\delta Vp = \pm 0.11$ km/s on average. The uncertainty propagation in the extrapolated velocities at zero



confining pressure ($\delta Vp0$) was based on the linear regression function that minimized the sum of the squared errors of the prediction. Subsequently, we used the error of the regression coefficients (i.e., slope and intercept) to derive the uncertainty

for the anisotropy coefficient (AVp) using the same approach as for the measured velocities. The uncertainty of the P-wave anisotropy was below 1 % for all samples. Moreover, our error analysis has shown, that the bulk uncertainty of the measured seismic velocities is dominated by the pick accuracy of the pulse arrivals, while errors in the plug length have only a minor impact.

## 2.2 Multi-sensor core log P-wave velocity

Seismic P-wave velocity was measured every 5 cm on the 2.5 km COSC-1 cores using a multi-sensor core logger (MSCL, type Geotek MSCL-S). The P-wave sensor setup comprised two signal transducers mounted on opposite sides, perpendicular to the core axis. The upper was a motor-driven piston transmitter, while the lower used a spring-loaded acoustic rolling contact (ARC), constantly pushed against the measured cores.

At each measuring position, a 230 kHz P-wave pulse was sent from the transmitter through the core and was recorded by
the receiver. The recorded signal was pre-amplified, digitized, and sent to the acquisition software at a sampling frequency of 12.5 MHz. The first-arrival times were determined using a threshold method providing a user-defined threshold level and time delay. This system automatically determined the first excursion above the given threshold, after the delay. The total travel time (TOT) was then taken at the first zero-crossing. During the acquisition procedure, the voltage level (amplitude), threshold level, and time delay were adjusted accordingly. To provide a good coupling between the transducers and the core, the core
surface was wetted before the measurement.

Furthermore, a calibration was carried out to determine the P-wave travel time offset (PTO), which accounts for the accumulated travel time delays through the transducer system (transducer faces, rubber plates, etc.). The actual travel time (TT) was derived by subtracting the PTO from total measured travel time, i.e., TT = TOT – PTO.

The calibration based on travel times measured on whole-round and half-split POM (*Polyoxymethylen*) cylinders of varying
diameters (60 to 120 mm), which were plotted and extrapolated by a best linear fit. The P-wave velocities were calculated using the corrected travel time TT and the nominal core diameter (D = 61 and 47.6 mm), by Vp = D / (TOT – PTO) = D / TT. No additional temperature correction was applied because the room temperature inside the laboratory was kept constant at about 21 °C.

## 2.1 Downhole sonic logging

A downhole sonic measurement was carried out as part of a complete downhole logging campaign in 2014, about one month after the drilling of the COSC-1 borehole was completed. The original data set is published by Lorenz et al. (2015b, 2019) and comprises, amongst others, a density log and 3D core scan images. Downhole sonic velocities were continuously logged every 0.1 m by the Operational Support Group (OSG) of the ICDP using a standard full-waveform slimhole sonde (Antares, Germany). Transit times were calculated from first arrival times, which first were picked automatically and then





refined manually. The seismic P-wave velocities were calculated using the sonde's receiver spacing (0.5 m) divided by the transit time.

Based on refracted waves propagating along the borehole wall, the sonic log represents a vertical average velocity of the near-well vicinity over the receiver spacing. The investigation depth and resolved rock volume generally depend on the formation velocities and frequencies, and can be estimated by three times the dominant wavelength (Serra, 1984). The frequency range of the recorded sonic traces lies in the order of a few kilohertz (Fig. 6). For the given sonde geometry and nominal transmitter frequency (20 kHz), the depth of investigation is about 0.75 to 1 m. Thus, downhole logging velocities provide a good approximation of the in situ seismic velocities, but can still be affected by micro-fracturing caused by drilling or steeply-dipping natural fractures.

## 2.2 Zero-offset vertical seismic profile

A zero-offset vertical seismic profile (ZO-VSP) was acquired in the COSC-1 borehole, as part of a comprehensive post-drilling seismic survey to image the SNC and its underlying formations (Krauß, 2017; Krauß et al., 2015). In our study, we used the P-wave velocities that were calculated by the first-arrival times of the consecutive downhole receiver stations. The receiver spacing was 2 m and the first-arrival times were smoothed by a 15-point moving average (equals 30 m interval) to account for small travel-time variations.

For a zero-offset (rig-source) VSP, the direct P-wave ideally propagates downwards from the surface, parallel to the borehole. Thus, being relatively unaffected by the borehole itself, borehole-seismic velocities provide a good approximation of the vertical in situ seismic velocity of the borehole vicinity. The calculated, so called interval velocity represents the constant velocity of the seismic wave travelling through a rock layer with a given interval thickness, which is defined by the applied receiver spacing (2 m). The distance over which the velocity was averaged was about 4 times larger than in the downhole sonic log.

The signal frequencies ranged between 80 and 100 Hz (Fig. 6). The measurement scale was mainly dictated by the horizontal and vertical resolution. The former can be approximated by the first Fresnel zone of the dominant seismic wavelength. For the COSC-1 borehole range (0-2500 m), this yielded an average horizontal resolution of about 300 m ($\lambda = 75$ m, $v_{const.} = 6$ km/s). In contrast, the vertical resolution was about 20 m, assuming one quarter of the dominant wavelength (best case).

## 3 Results

### 3.1 Laboratory data

The laboratory intrinsic seismic velocities lie between 5.9 and 6.9 km/s showing generally little scattering (std = 0.3 km/s) throughout all samples (Table 2). The slowest velocities occur always along the z-axis, thus perpendicular to the foliation plane. Highest velocities occur in the foliation plane, i.e., along the x- and y-axes. The intrinsic seismic anisotropy exhibits a





strong variation between 1 and 26 %, with an average error of 0.4 %. The average seismic anisotropy for all 16 samples is about 10 % (Table 2).

Velocities calculated at lithostatic pressure show values between 4.8 and 6.8 km/s (std = 0.5 km/s). Velocities calculated at atmospheric pressure are significantly lower, ranging from 1.5 to 5.7 km/s on average (std = 1.3 km/s). The measured sample
densities vary between 2.7 and 3.1 g/cm³, with the highest densities observed for the amphibole-rich (mafic) samples (149-4, 193-2, 556-2, 631-1, and 661-3).

The mean intrinsic P-wave velocities (Vp0) were derived from the arithmetic mean of the three core plugs and represent the most general case excluding any directional or structural effects and mainly account for the compositional effects (Fig. 7). Amphibole-rich (mafic) rock samples have velocities ranging between 6.5 and 6.9 km/s, whereas all other more felsic rock
samples including the felsic gneisses, mica schists, and metasandstone, are characterized by a Vp0 between 6.0 and 6.4 km/s. The lowest Vp0 can be associated with the felsic gneiss samples, while the mica-rich schists show slightly higher Vp0. Moreover, both metasandstone (sample 664-2) and the carbonate-rich gneiss (sample 243-2) show very similar Vp0 as for the mica schists (e.g., samples 641-5, 651-5).

The seismic P-wave anisotropy (AVp) changes with increasing depth. This provides a simplified anisotropy-depth profile
along the COSC-1 borehole (Fig. 8). The uppermost about 600 m show medium anisotropies (<10 %) and low values (< 5 %) between 750 and 1500 m. Between 1600 and 1900 m, we observe the highest anisotropy effect with values up to 25 %, which decreases again, further below.

Comparing the different velocity distributions, the velocities at atmospheric pressure (VpAP) show very strong scattering and generally low values. These agree well with the velocities measured on core under similar pressure conditions (Fig. 9).
Velocities at lithostatic pressure (VpLP), in contrast, follow the in situ velocities measured downhole by the sonic tool and zero-offset VSP. On average, the intrinsic velocities (Vp0) are slightly higher than those calculated under lithostatic pressure.

### 3.1    Lab, core, and log data integration

Downhole sonic log and VSP show a good correlation, while VSP velocities have a lower resolution caused by the averaging. The raw core velocities show a strong scattering with lower velocities on average (Fig. 9).
There is neither a clear correlation between core and downhole velocities nor an observable static offset with depth (Fig. 10). There are places where the core-derived Vp increases while the downhole-measured Vp decreases, for example in the lowermost 200 m of the borehole. Moreover, we can observe core velocities that closely approach or even exceed the downhole velocities (430 to 780 m; 1700 to 2000 m), while at other depth intervals (620 m, 1640 m, and 1800 m), core and downhole Vp mismatch significantly. Between 160 and 180 m, we see a strong decrease in downhole P-wave velocities, which are likely
related to a karstic unit previously recognized by a very high secondary porosity (Lorenz et al., 2015b). Here, the core and downhole velocities show a good agreement.

From about 430 to 780 m we observe several peaks in the core velocities, which also correlate with peaks in the downhole velocity and density logs. Between 780 and 1900 m the core velocities gradually increase and they are accompanied by several





smaller, less pronounced peaks, which often match with peaks in the downhole velocity and density logs (e.g., between 900
and 1000 m). From about 1900 m, down to about 2350 m, the core velocities tend to decrease, before they increase again
abruptly and clearly approach the downhole velocities.

In general, the superimposed mean sample velocities (intrinsic, atmospheric, and lithostatic) correlate well with the
associated core and downhole velocities. The sample densities match almost perfectly the core and downhole density
measurements. While the mean intrinsic velocities agree mostly with the downhole seismic velocities, they are slightly higher
in some depth intervals (e.g., samples 193-2, 631-1), possibly due to anisotropy effects. The velocities calculated at lithostatic
pressure generally follow the downhole velocities showing slightly lower values in the uppermost about 1600 m. Below, they
are very similar to the intrinsic velocities (Fig. 10, markers are partly overlapped).

The velocities calculated at atmospheric pressure match the core velocities except for the samples 361-2, 556-2, 631-1, and
691-1. For samples 243-2 and 487-1 the mean velocities at atmospheric pressure are exceptionally low (1.7 and 1.5 km/s),
thus, outside the displayed value range (see also Table 2).

At about 1750 and 1880 m, core and downhole velocities agree very well and the sample velocities (569-2 and 593-4) at
atmospheric pressure are close to that at lithostatic pressure. Fracture mapping indicates a higher amount of low-angle fractures
at these depths (Wenning et al., 2017). Moreover, these samples show the highest anisotropy values of all samples. We cannot
observe any direct correlation of the foliation dips and the velocity data.

**3.2    Comparison of velocity data at core scale**

We conducted a detailed analysis of the measured seismic velocities at core scale (cm to mm), for six selected core sections
(Fig. 11 A-F), which represent characteristic lithological units with respect to their seismic properties. We compared the
measured core and downhole velocities with the laboratory results and correlated them with the unrolled 360° core scans of
each selected core section. Missing core data are caused by samples taken previously to the core measurements.

Section 106-1 (Fig. 11 A) is a migmatite unit that is characterized by an alteration of darker restite bands and leucocratic
melts (chemically very similar to felsic gneiss). Both core and downhole velocities are relatively stable showing an average
difference of about 1.6 km/s. Sample velocities calculated at atmospheric conditions agree with the very low core Vp. The
lithostatic velocity, however, is lower than those logged downhole, whereas the intrinsic velocities are higher. This indicates
a strong effect of microcracks with poor orientation. The intrinsic seismic anisotropy is comparably low (< 5 %). Similar
results were obtained for a gneiss sample (sample 143-1, not shown), which have slightly higher core velocities (ca. 4.5 km/s)
but similar downhole Vp (ca. 5.6 km/s).

Section 193-2 (Fig. 11 B) contains amphibolite where core and downhole velocities match well. We observe similar results
for the metagabbro sections (e.g., sample 149-4, not shown), which are chemically almost equivalent. A fracture in the core
section can be clearly identified by the core Vp. The velocity at lithostatic pressure matches almost perfectly the downhole
velocity. The very low velocities at atmospheric pressure of the laboratory samples could be an artefact of an improper velocity-
pressure relation for this sample. In comparison with the amphibolite sections, the metagabbro sections exhibit slightly higher





downhole velocities, which, however, still agree with velocities measured on core. In general, both units show very similar characteristics.

Section 361-2 (Fig. 11 C) is dominated by gneiss of felsic composition and shows similar characteristics as the uppermost
gneiss sections (e.g., 143-1, not shown) and slightly higher velocities, which could relate to an increase of mafic minerals such as amphibole. The core and downhole Vp differ strongly by up to 2 km/s, while the downhole velocities are around 6.2 km/s. A small amphibolite layer (<5 cm in thickness), can be well resolved by the core velocity. The core velocity of the surrounding gneiss unit corresponds well with the atmospheric velocity and the intrinsic velocity. It matches with the downhole measurements, which we interpret as an effect of closure of most microcracks at in situ lithostatic pressure.

Section 569-2 (Fig. 11 D) contains mostly mica schist. It exhibits the strongest anisotropy (15 to 20 %) of the cored rocks. The core velocities (ca. 5 to 5.5 km/s) generally agree with the downhole velocities (ca. 5.5 km/s), being only slightly lower. The horizontal velocities at atmospheric pressure (see labels in figure) are even higher than the downhole velocities and the z component of the velocity at lithostatic pressure. However, the core velocities are still lower than the downhole velocities.

The amphibole-bearing gneisses of section 661-3 (Fig. 11 E) have relatively low core velocities of about 4.8 km/s. In
contrast, the downhole velocities are very high (6.4 km/s), which agree well with the z-component of the velocity at lithostatic pressure. The velocities at atmospheric pressure scatter strongly around the core velocities. This suggests a strong effect of microcracks on the extracted core section.

The metasandstone section 664-2 (Fig. 11 F) is characterized by a very homogenous rock matrix and is predominately composed of quartz. The velocity anisotropy is very low (< 5 %). Both atmospheric and lithostatic velocity match well the
corresponding core and downhole measurements. The present velocity differences are likely caused by microcracks, as indicated by the sample velocities. The core velocities for this section are almost constant (4.5 to 4.8 km/s), slightly higher than those measured in the uppermost felsic gneisses (e.g., section 361-2 in Figure 11 c).

## 4    Discussion

### 4.1    Laboratory seismic properties

Our laboratory investigations show that not only composition but also structural characteristics of the COSC-1 cores have a strong impact on seismic properties. Pechnig et al. (1997) showed that the physical properties of metamorphic rocks can be classified by both structure and composition. We have shown that samples from mafic rocks have average velocities higher than 6.5 km/s and densities above 2.9 g/cm³, whereas felsic gneisses and mica schists show lower P-wave velocities and densities of 2.7 to 2.8 g/cm³ (Table 2, Fig. 7). These results fit well with the characteristics observed on felsic and mafic rocks
from the German Deep Drilling Program KTB (Bartetzko et al., 2005; Pechnig et al., 2005). We suggest that velocity contrasts mainly occur between the denser amphibole-rich units and the more felsic units including felsic gneisses, mica schists, and metasandstones.





Nevertheless, the mean intrinsic seismic velocity cannot clearly distinguish all investigated rock types probably because of very similar matrix velocities and rock compositions. Our results show that the P-wave seismic anisotropy provides additional

information about the structural characteristics, which qualitatively correlates with the degree of foliation. We observed the highest anisotropies (> 15 %) for the mica schists, which are characterized by a well-developed schistosity. In contrast, felsic gneisses and metasandstone samples showed anisotropies about 5 % or below. This suggests a strong structural dependence of the seismic velocities for the present rocks.

Based on the anisotropy depth profile (Fig. 8), we can distinguish four different zones, which correspond to major

lithological units: (1) medium-low AVp of alternating, very heterogeneous rock units (samples 106-193, 400-650 m), (2) very low AVp of felsic rocks with low schistosity (samples 243-2 to 487-1, 790 to 1500 m), (3) high AVp of mica-rich rocks with well-developed schistosity (samples 556-2 to 651-5, 1690 to 2220 m), and (4) low AVp of granofelsic quartz-feldspar-rich rocks (sample 664-2, >2280 m). The lowermost depths were also not well constrained, being covered by only two samples, one metasandstone and one mica schist. According to the core description, however, most of the deepest (>2200 m) rock units

are described as metasandstones with only a few layers of mica schist (Lorenz et al., 2015b). The here presented anisotropy-depth profile is limited in resolution by the low number of samples. Nevertheless, despite of large data gaps we are able to divide the borehole into structural units that are not detectable based on other seismic properties.

Rocks of the Seve Nappe Complex were subject to high- to ultrahigh pressure metamorphism (Arnbom, 1980; Klonowska et al., 2017; Majka et al., 2014), involving both structural and compositional changes of the protolith. Metamorphism may

affect differently the seismic properties depending on the p-T history. Generally, we assume an increase in seismic velocity with increasing metamorphism due to compaction and formation of denser minerals. On the other hand, seismic anisotropy at rock scale can either increase or decrease with increasing metamorphism due to crystallographic preferred orientation and dynamic recrystallization of constituent minerals under variable stress and temperature conditions (Bezacier et al., 2010; Falus et al., 2011; Keppler et al., 2017). We observed that for the upper 1.6 km of the COSC-1 borehole, the seismic anisotropy is

lower for the high-grade gneisses and amphibolites while at greater depths (>1.6 km) high anisotropy is associated with lower-grade mica schists (Fig. 10).

Laboratory studies (e.g., Babuska and Cara, 1991; Kern and Wenk, 1990; Shaocheng and Mainprice, 1988) have shown that seismic anisotropy can be affected by the degree of deformation, such as associated with high-strain rates and the mylonitization of rocks. Because of the associated lineation or stretching of minerals, this can favor an increase in the seismic

anisotropy, as we observe, for example, for the amphibole-rich gneiss sample (sample 661-3, Fig. 8). However, from the core or log velocities alone, we are not able to find strong evidence for a shear zone interface or zones of mylonitic deformation. Better constraints of the effects of tectonic deformation at the sample scale require additional analysis of the microstructure and related anisotropy.



### 4.2 Seismic velocities under laboratory and in situ condition

We used sample velocities measured at increasing confining pressure using a hydrostatic pressure vessel (Figure 3) to simulate velocities measured under atmospheric and downhole conditions. Based on pressure-velocity curves, we calculated velocities that represent either intrinsic, core, or downhole logging conditions (Fig. 9).

    For the uppermost samples, we observed higher intrinsic velocities than velocities calculated at their lithostatic pressure (Fig. 10). This is counter-intuitive because we would assume that the velocities calculated at lithostatic (i.e., in situ) pressure

are higher or at least similar to those calculated at zero confining pressure. If this is not the case, the calculated lithostatic pressure is not high enough to exceed the non-linear (crack-related) part of the velocity-pressure relation. This implies that the in situ velocities for these rocks are stronger influenced by fractures or microcracks than the velocities of samples at greater depth in the borehole.

    Both core velocity and core density, which we used to calculate the lithostatic pressure, were measured under dry-rock

conditions. If compared with in situ measurements, the effect of (partial) saturation could explain why velocities are lower than under in situ conditions (e.g., Kingdon et al., 1998). However, Fountain (1976) showed that this effect should be negligible in crystalline rocks with low porosity such as those in this study. Very similar density values from the core and downhole measurements (Fig. 4) supports that water saturation does not have a big impact.

    To simulate velocities under in situ pressure conditions, we calculated velocities at their lithostatic pressure (Fig. 4). This

assumes that the principal stresses are equal in all directions and determined only by the overlying rock masses (e.g., Zang and Stephansson, 2010). But the in situ stress field can be more complicated due to tectonic processes such as ridge push, post-glacial relief, or mantle-driven stress. For the COSC-1 area, the in situ stress anisotropy is low (Wenning et al., 2017). Thus, we assume that lithostatic pressure is a good approximation for the in situ pressure conditions. This is further confirmed by the good correlation between the velocities calculated at lithostatic pressure and the downhole logging velocities (Fig. 9).

Low mean velocities at atmospheric pressure for some of the samples (Fig. 10: 361-2, 556-2, 631-1, and 691-1) could result from very low velocity in either one of the associated core plugs. This may result from insufficient data coverage of the low-pressure part of the velocity-pressure relation (Figure 5) causing wrong data extrapolation. Another source for such misfits is the different pressure relation used for the samples investigated by Wenning et al. (2016). This was based on the velocity pressure relationship proposed by Wepfer and Christensen (1991). This empirical relationship is adequate at higher pressures

but not for zero confining pressure. Thus, velocities calculated at atmospheric pressure are generally too low.

    We infer that the observed difference between intrinsic velocity and those calculated at lithostatic pressure is mainly caused by microcracks induced by anisotropic stress relaxation after coring downhole (e.g., Wolter and Berckhemer, 1989). Due to the insufficient closure of microcracks the onset of the linear part of the velocity-pressure curve is shifted to higher (p > 70 MPa) confining pressures whereas the calculated lithostatic pressure is located in the non-linear part. The fact that this effect

mainly occurs in the uppermost, less schistose samples (e.g., 143-1, 361-2) suggests that these samples are more affected by



microcracks and that in the schistose samples microcracks are more aligned and therefore can close faster under increasing pressure.

Our results suggest that the intrinsic seismic velocities are a good representation of the in situ seismic velocities as measured by downhole logging. Although the velocities calculated at lithostatic pressure generally agree with the downhole velocities,
the insufficient closure of microcracks leads to slightly too low values for the felsic gneiss units.

## 4.3     Characteristics of core and downhole logging Vp measurements

Core and downhole velocity measurements are subject to different scales, sensor setup, and environmental conditions (Fig. 2). Other studies have shown that differences in seismic properties are generally due to depressurization and formation of microcracks after the core extraction (e.g., Wolter and Berckhemer, 1989; Zang et al., 1989). Especially in sedimentary rocks,
the mechanical rebound of pore spaces due to decompression is a primary correction factor when comparing core to in situ data (Urmos et al., 1993). For crystalline, metamorphic rocks, the effect of volume expansion is relatively small. Microcracks can be either randomly distributed or show a preferred orientation relative to the rock microstructure or to the stress field around the borehole (Dresen and Guéguen, 2004; Nur and Simmons, 1969). Our simulation of velocities under crack-related and crack-free conditions (Figure 5) indicate a strong influence of microcracks and a significant crack-induced anisotropy for
certain rock samples (e.g., Fig. 11 A). This suggests that velocities measured on cores at atmospheric pressure are strongly affected by microcracking.

Fig. 10 illustrates the significant differences between the core and downhole seismic velocities at several depths. The discrepancy between the core and downhole logs can have different reasons. As discussed above, the decompression of the cores causes the formation of microcracks (asymmetric strain relaxation). With respect to the sample lithology (Table 2), we
observe the strongest mismatch and lowest core velocities for the gneiss units. For the metasandstones, mica schists and mafic rocks the mismatch is comparably low and core velocities are increased. Especially, between about 450 and 800 m, core and downhole velocities matched very well, which coincides with mafic lithologies, i.e., metagabbros, amphibolites. These have higher densities, but they are probably also less affected by microcracks. We infer that microcracks impact more on the core velocities of the felsic gneisses than on other lithologies. Despite of the general mismatch in the core and downhole velocities,
the core velocities resolve both large-scale and small-scale (less than about 5 cm) lithological changes that are related to mafic rocks (e.g., Fig. 11 C).

Other effects on the measured core velocities can be related to structural characteristics like the presence of dipping foliation, natural fractures, and grain boundary orientations. We show that MSCL measurements are sensitive to fractures in the rock resulting in too low velocities (see Fig. 11 B, D). Beside some naturally occurring fractures (see fracture column in
Fig. 10; Wenning et al., 2017), most fractures are due to core handling and there are of course those that occur inevitably between each core section. In total, natural fractures and core section transition zones account for about 10 % of the raw data set suggesting that they are not the reason for the general mismatch. This is supported by the smoothed core velocities, where all outliers were removed (Fig. 10, smoothed core Vp profile).





The core data, unlike the downhole logging data, were acquired perpendicular to the core axis and, thus, parallel or sub-
parallel to the metamorphic foliation (Fig. 2). In the presence of a well-developed foliation this would cause higher core
velocities in horizontal direction, parallel to the foliation, as shown by our laboratory results where the fastest velocities always
occur parallel to the foliation plane. However, at core scale and atmospheric pressure, this effect is less eminent and is only
observed where a strong schistosity is present (Fig. 11 D).

The core and borehole velocity measurements were carried out using different frequencies ranging from a few 100 Hz for
the VSP data to up to 1 MHz for the sample analyses (Fig. 6). In general, P-wave velocity and anisotropy dispersion may occur
in porous and fractured rocks (Galvin and Gurevich, 2015; Thomsen, 1995). In absence of a fluid-saturated, equant porosity
as in the case of mostly crystalline metamorphic rocks, P-wave dispersion can generally be neglected. Moreover, the different
frequency scales are closely linked to the investigated rock volume, i.e., with increasing signal frequency the investigated rock
volume generally decreases.

The seismic properties show considerable changes between each data set (Fig. 12). Furthermore, there is no strong
correlation between the lithological core description and the seismic velocities, which might be related to the very variable
core lithology. Some characteristic zones in the velocity profiles, however, can be linked to the more general trend in the
lithology. Most eminent is the transition between the mafic, amphibole-rich rocks and the felsic gneisses at around 750 to 800
m. Overlying mafic units can be traced across all scales indicated by strong velocity contrasts and peaks between 400 and 800
m due to the layering of mafic and felsic rock units. In the MSCL data we can associate the low velocities with felsic gneisses,
whereas the mica schists and metasandstones show slightly increased core velocities. While the differentiation of mica schists
and gneisses, is very difficult due to the similar seismic velocities, high anisotropy values give indications for a unit dominated
by mica schists, which also agrees with the lithological core description. This implies that the core-derived velocities cannot
easily be used for core log seismic integration, both because they resolve the velocities at much greater resolution and because
(in case of the MSCL data) they are too low due to decompression.

Core and log velocities suggest that velocity contrasts in the Lower Seve Nappe are mainly related to the mafic rock units,
which are can be associated with boudinaged amphibolites (Hedin et al., 2016) or dolerite intrusions (Juhlin, 1990). We
conclude that these mafic units predominately occur in the uppermost 800 m, in thick bands of 10 to 80 m, exhibiting
intermediate intrinsic anisotropies of up to 8 %. Potentially the core velocities can be used to derive a high-resolution
reflectivity series with a high contrast level (dynamic range), which could help to better localize the origins of seismic
reflections related to mafic units such as amphibolites and metagabbros.

## 5   Conclusion

Integrating seismic properties from laboratory samples, core and downhole data helps to better define and distinguish
seismic characteristics of the COSC-1 lithology. Our comparison of seismic velocity measurements across multiple scales,
from laboratory to field, show that seismic P-wave velocities measured on cores can partly resolve in situ lithological



variations. These are, however, not only influenced by compositional changes but also overprinted by metamorphic foliation and microstructures.

The investigated seismic properties allow to distinguish between at least three characteristic lithological units of the COSC-1 borehole: amphibolite/metagabbros, mica schists, and felsic gneisses. Although core velocities are affected by microcracks, they are able to resolve small scale features such as thin mafic layers (<5 cm) or sub-horizontal fractures. We were able to identify mafic units such as amphibolite by peaks in both core and downhole data, which can be used for lithological classification. Less prominent velocity contrasts but outstanding anisotropy values were observed for mica schists and metasandstones. Core and downhole velocity contrasts can be attributed to the transition from mafic to other lithologies, which dominates the uppermost borehole sections. Where core and downhole velocities show discrepancies this can be ascribed to microcracks induced by the coring process (strain relief of the core), and this occurs more frequently in the gneissic lithologies.

## 6    Acknowledgement

Research funded by the Deutsche Forschungsgemeinschaft (DFG, German Research Foundation) – Project no. 339380967 (SPP-1006). The COSC-1 borehole was funded by the Swedish Scientific Drilling Program (SSDP) through the Swedish Research Council (VR-2013-94) and the International Continental Scientific Drilling Program (ICDP). We thank Bjarne Almqvist, Henning Lorenz, Iwona Klonowska, and Jaroslaw Majka for constructive and fruitful discussions during manuscript preparation. We are grateful to Claudio Madonna and Quinn Wenning for providing their support and advice during the preparation and measurements of the samples at the Rock Deformation Laboratory of the ETH Zurich. We would like to thank Ulli Raschke for his assistance and efforts during core logging with the multi-sensor core logger at the BGR core repository in Berlin-Spandau, and Friederike Klos for providing logistical support for the sample material.

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

## 8   Data availability

All core and downhole logging data are available through the ICDP data repository (cosc.icdp-online.org) and are referenced in the text with the respective data publication. The newly acquired data will be curated by the ICDP data repository (http://doi.org/10.5880/ICDP.5054.002).



## 9     Sample availability

Each of the 16 core samples has an IGSN (see Table A1). The samples are stored at the individual institutes and can be requested from the authors.

## 10     Author contribution

FK and SP carried out the core measurements. FK and AZ conducted the laboratory measurements and analysis. FK
prepared the manuscript with contributions from all co-authors. SP and CB are responsible for conceptualization and funding acquisition. All authors worked on the manuscript.

## 11     Competing interests

The authors declare that they have no conflict of interest.





**12 Figures**

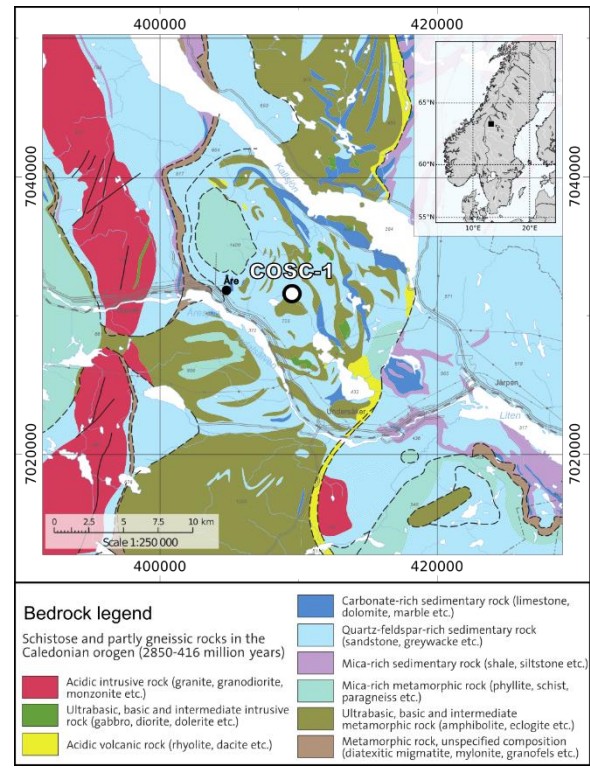

**Figure 1.** Geological map of the Åre-Järpen area in western Scandinavia (see inset) showing the COSC-1 borehole location (circle marker). Bedrock map from Geological Survey of Sweden (SGU).

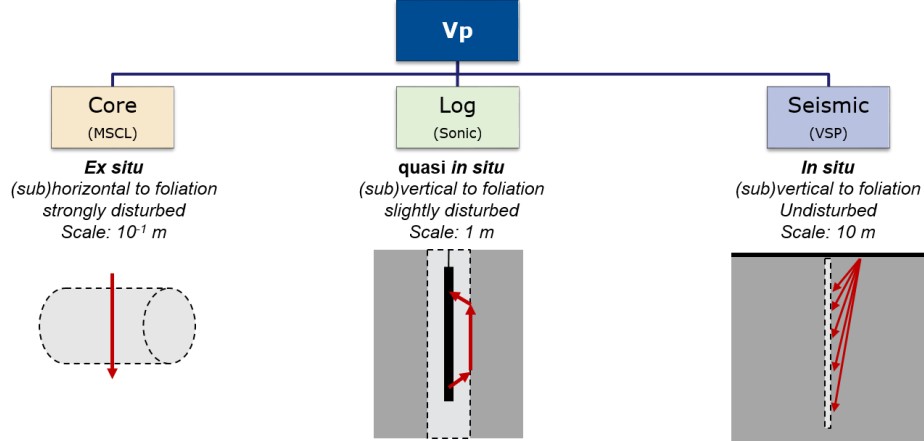


**Figure 2.** Schematic depicting the different environmental and measurement conditions of the core, log, and borehole seismic (VSP) P-wave velocity (Vp) measurement.



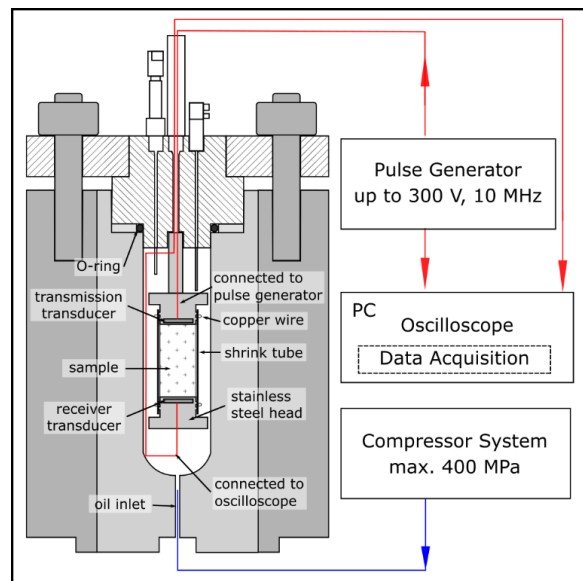

**Figure 3.** Schematic of the experimental setup used to determine the P-wave velocities under confining pressure. This setup comprises a pressure vessel with a sample chamber, pulse generator, compressor control, and PC-based acquisition unit. Placed inside the oil-filled pressure chamber there is the sample assembly (Barblan, 1990).

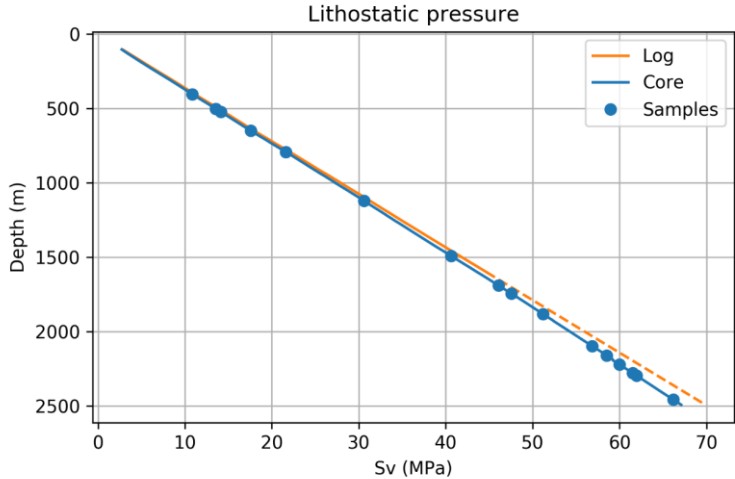

**Figure 4.** Lithostatic pressure curve of the COSC-1 borehole derived from core and downhole density measurements. The values for the 16 core samples are marked accordingly. The dashed line was linearly extrapolated since downhole density was only logged down to about 1.6 km.


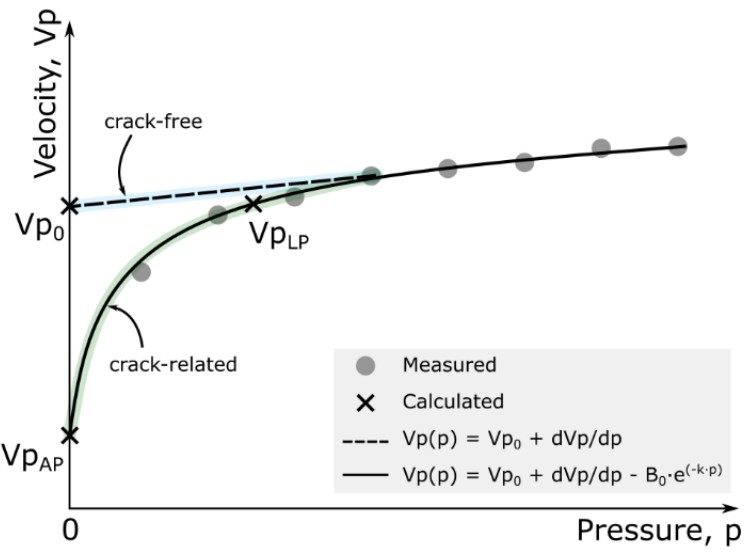

**Figure 5.** Velocity-pressure relation depicting the measured and calculated seismic velocity Vp as a function of confining pressure p. Based on the best curve fit for the linear (dashed line) and non-linear part (solid line), velocities can be calculated under different environmental conditions: Vp0 – intrinsic velocity, VpAP – atmospheric pressure, VpLP – lithostatic pressure (cf. Table 1).

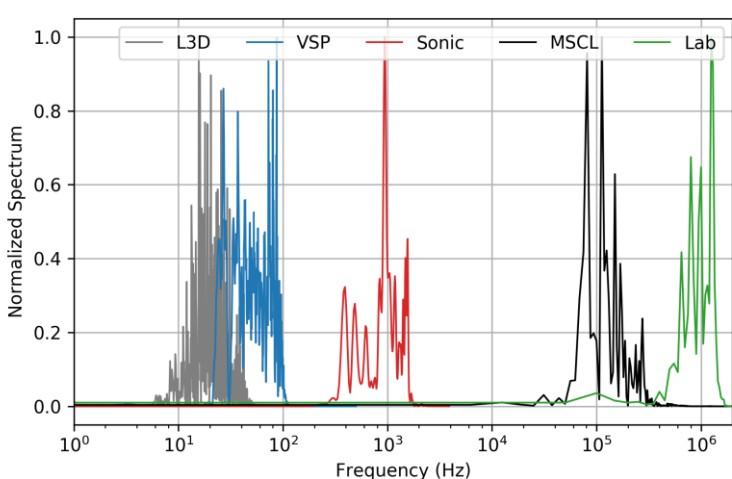

**Figure 6.** Comparison of frequency spectra of seismic measurements across multiple scales from limited 3D surface seismic (L3D), borehole seismic (VSP), downhole log (sonic), core measurement (MSCL), and laboratory samples (Lab). The downhole-related spectra are calculated from a single seismic trace and sonic waveform extracted from approximately the same downhole depth of about 500 m.





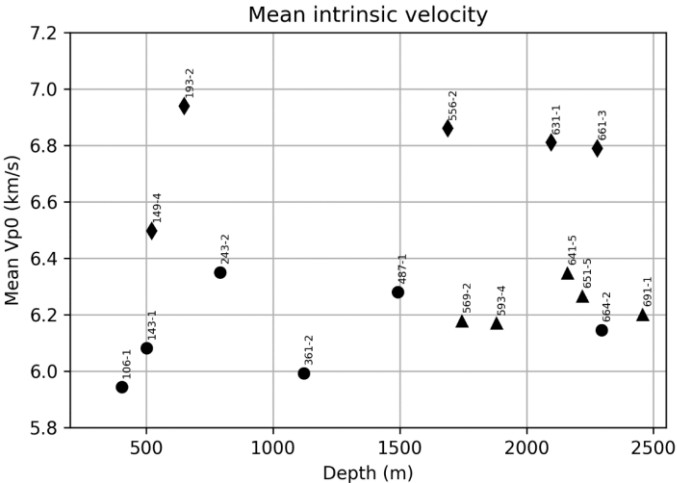

**Figure 7.** Mean intrinsic velocity measured on 16 core samples from the COSC-1 borehole plotted at the respective sample depth. Markers correspond to simplified lithological classes such as mafic amphibole-rich units (♦), felsic gneisses/metasandstones (●), and mica schist (▲). Note that the highest values occur for the mafic lithologies.

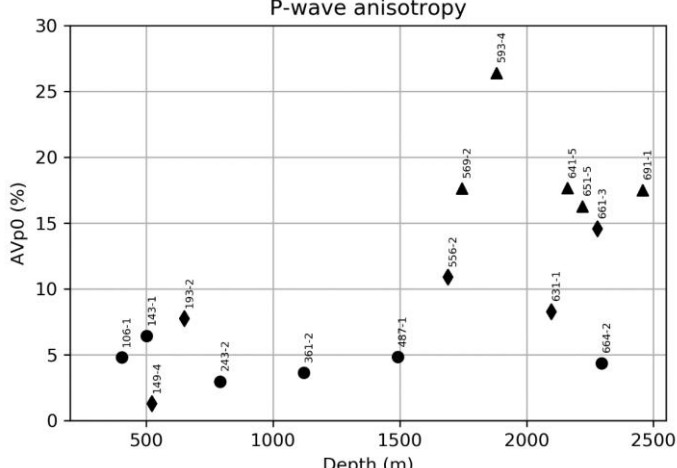

**Figure 8.** Seismic anisotropy measured on 16 rock samples from the COSC-1 borehole plotted at the respective sample depth. Markers correspond to simplified lithological classes such as mafic amphibole-rich units (♦), felsic gneisses/metasandstones (●), and mica schist (▲).





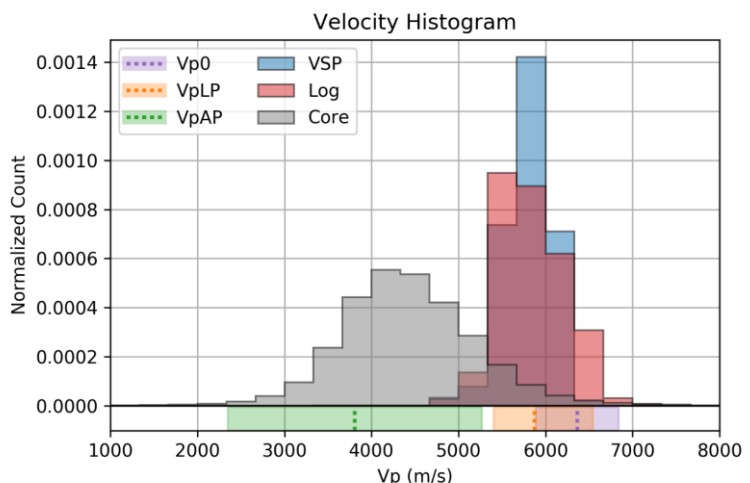

**Figure 9.** Distribution of P-wave velocities derived from laboratory samples (Vp0, VpLP, VpAp), core measurements (Core), downhole logging (Log), and borehole seismic (VSP). The sample velocities are displayed below the histograms indicating the mean and standard deviation of the 16 laboratory core samples.





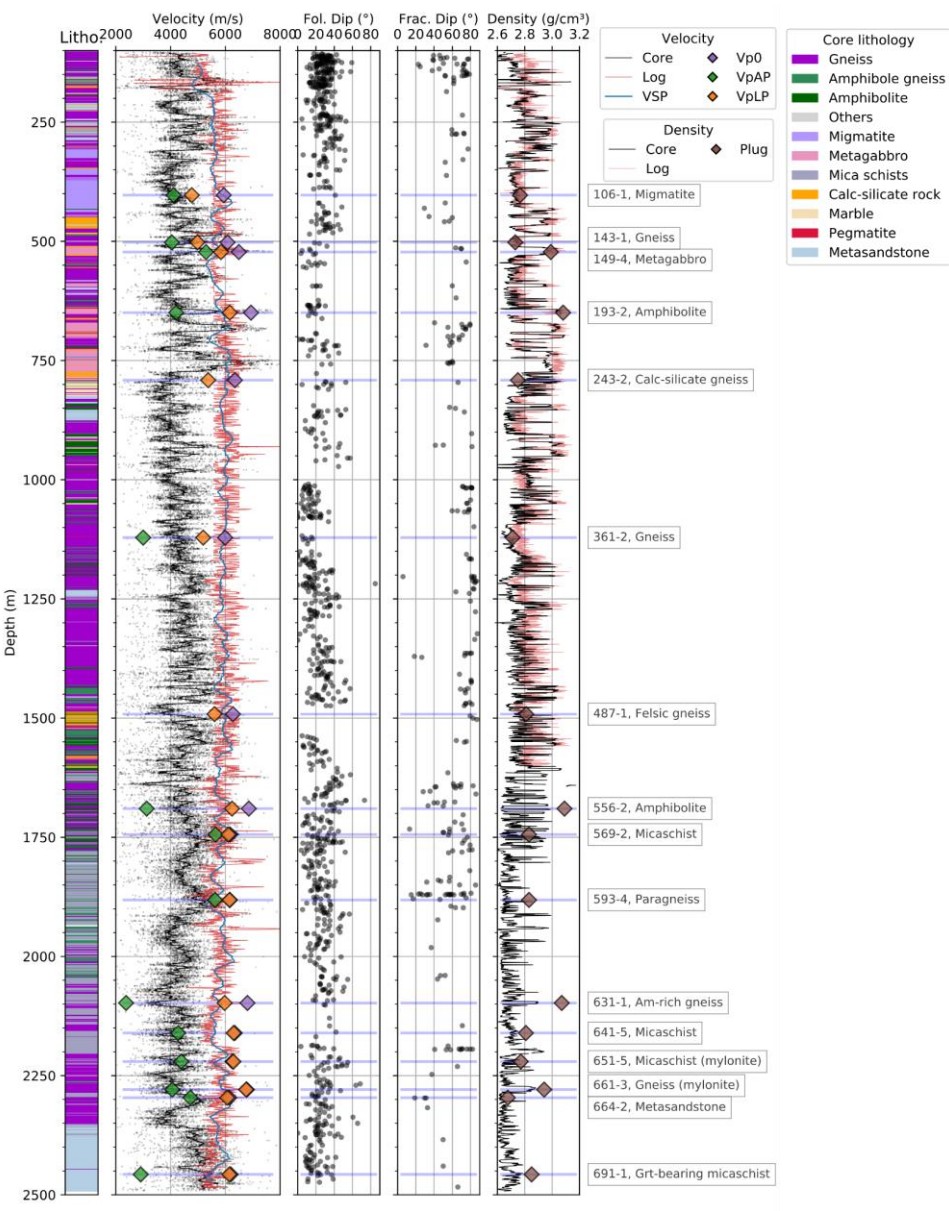

**Figure 10.** Core-log data integration of the seismic properties alongside with the fracture and foliation dip in the COSC-1 borehole. The sample velocity and density data are superimposed on the respective log panels. The lithology is based on the COSC-1 lithological description of the core (modified after Lorenz et al., 2015b).





**Figure 11.** Comparison of P-wave velocities across different scales. A-F) Laboratory sample, core measurements and downhole sonic velocities are shown next to the unrolled, true-color core scans. The colored bars represent the location of the three plug locations (for colors refer to previous figures); black dots: core velocity from MSCL, red dashed line: downhole sonic velocity. The blue and red lines on the core images are common practice to indicate the top and bottom of the core.





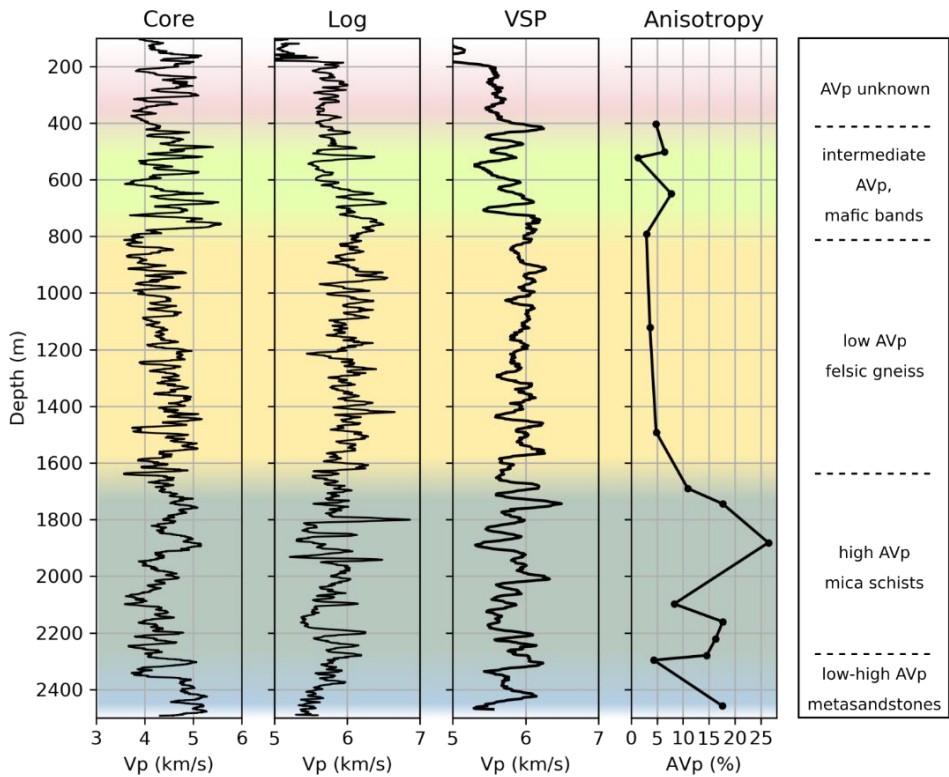

**Figure 12.** Up-scaled core and log and borehole seismic (VSP) velocities together with the laboratory-derived anisotropy depth-profile. Core
690    and downhole logging velocities are smoothed over a 10 m average. The anisotropy profile is linearly interpolated (see also Fig. 7). The
colored sections highlight characteristic velocity zones based on the resultant data sets shown here.





## 13 Tables

**Table 1.** Overview of velocity nomenclature used in this study. The laboratory measurements were carried out on three mutually
695  perpendicular core plugs. See text for details.

| Velocity | Type of measurement | Direction of measurement | Description |
|---|---|---|---|
| Vp0 | Lab | triaxial | Intrinsic P-wave velocity based on laboratory measurements on core plugs |
| VpAP | Lab | triaxial | P-wave velocity calculated at atmospheric pressure (p = 0.1 MPa) based on laboratory measurements on core plugs |
| VpLP | Lab | triaxial | P-wave velocity calculated at lithostatic pressure (p = Sv(z)) based on laboratory measurements on core plugs |
| Vpcore | MSCL | perpendicular to core axis | P-wave velocity continuously measured on whole cores using a multi-sensor core logger |
| Vplog | Downhole sonic | parallel to borehole axis | P-wave velocity continuously logged downhole using a short-spacing sonic sonde |
| VpVSP | Borehole seismic | parallel to borehole axis | P-wave velocities measured downhole using a zero-offset vertical seismic profile |



**Table 2.** Laboratory data of 16 rock samples from the COSC-1 borehole. Sv - Lithostatic pressure derived from core density.

| Sample | Core Lithology | Depth (m) | Density (g/cm³) | Sv (MPa) | Intrinsic velocity Vp0 (km/s) | | | | Intrinsic anisotropy (%) | | Extrinsic velocity VpAP (km/s) | | | | Lithostatic velocity VpLP (km/s) | | | |
|---|---|---|---|---|---|---|---|---|---|---|---|---|---|---|---|---|---|---|
| | | | | | x | y | z | Mean | AVp0 | Error | x | y | z | Mean | x | y | z | Mean |
| 106-1 | Migmatite | 403.09 | 2.76 | 10.83 | 5.93 | 6.10 | 5.81 | 5.94 | 4.79 | 0.35 | 4.08 | 4.45 | 3.80 | 4.11 | 4.72 | 5.05 | 4.56 | 4.78 |
| 143-1 | Gneiss | 502.10 | 2.73 | 13.56 | 6.09 | 6.28 | 5.87 | 6.08 | 6.44 | 0.62 | 3.91 | 4.51 | 3.73 | 4.05 | 4.94 | 5.28 | 4.73 | 4.98 |
| 149-4 | Metagabbro | 522.41 | 2.99 | 14.13 | 6.53 | 6.52 | 6.44 | 6.50 | 1.29 | 0.67 | 6.38 | 5.83 | 3.68 | 5.30 | 6.44 | 6.06 | 5.02 | 5.84 |
| 193-2* | Amphibolite | 649.59 | 3.08 | 17.58 | 7.23 | 6.93 | 6.67 | 6.94 | 7.75 | NA | 4.33 | 4.03 | 4.27 | 4.21 | 6.42 | 6.13 | 5.95 | 6.17 |
| 243-2* | Calc-silicate gneiss | 791.22 | 2.75 | 21.59 | 6.44 | 6.37 | 6.25 | 6.35 | 2.95 | NA | 1.50 | 2.40 | 1.04 | 1.65 | 5.40 | 5.55 | 5.18 | 5.38 |
| 361-2 | Gneiss | 1121.48 | 2.71 | 30.61 | 5.93 | 6.13 | 5.91 | 5.99 | 3.63 | 0.92 | 3.18 | 3.43 | 2.40 | 3.00 | 5.18 | 5.44 | 4.96 | 5.19 |
| 487-1* | Felsic gneiss | 1491.72 | 2.81 | 40.63 | 6.42 | 6.31 | 6.11 | 6.28 | 4.83 | NA | 1.66 | 1.40 | 1.40 | 1.49 | 5.75 | 5.65 | 5.42 | 5.61 |
| 556-2* | Amphibolite | 1689.94 | 3.09 | 46.08 | 7.25 | 6.88 | 6.46 | 6.86 | 10.90 | NA | 4.73 | 2.65 | 2.02 | 3.13 | 6.81 | 6.36 | 5.57 | 6.25 |
| 569-2 | Mica schist | 1744.27 | 2.83 | 47.55 | 6.67 | 6.36 | 5.50 | 6.18 | 17.63 | 0.18 | 6.12 | 5.94 | 4.89 | 5.65 | 6.60 | 6.32 | 5.43 | 6.12 |
| 593-4 | Paragneiss | 1881.51 | 2.83 | 51.18 | 6.90 | 6.52 | 5.08 | 6.17 | 26.41 | 0.17 | 6.50 | 6.08 | 4.29 | 5.63 | 6.89 | 6.47 | 5.10 | 6.16 |
| 631-1* | Am-rich gneiss | 2097.65 | 3.07 | 56.87 | 7.12 | 6.78 | 6.53 | 6.81 | 8.29 | NA | 3.22 | 2.13 | 1.78 | 2.38 | 6.27 | 6.04 | 5.62 | 5.98 |
| 641-5 | Mica schist | 2160.64 | 2.81 | 58.53 | 6.92 | 6.41 | 5.70 | 6.35 | 17.65 | 0.56 | 4.37 | 4.93 | 3.53 | 4.28 | 6.98 | 6.36 | 5.57 | 6.31 |
| 651-5 | Mica schist (mylonite) | 2220.52 | 2.77 | 59.97 | 6.74 | 6.41 | 5.64 | 6.27 | 16.27 | 0.27 | 5.58 | 5.08 | 2.51 | 4.39 | 6.76 | 6.46 | 5.66 | 6.30 |
| 661-3 | Am-rich gneiss (mylonite) | 2279.31 | 2.94 | 61.53 | 7.32 | 6.79 | 6.26 | 6.79 | 14.56 | 0.34 | 2.74 | 5.94 | 3.53 | 4.07 | 7.35 | 6.73 | 6.21 | 6.76 |
| 664-2 | Metasandstone | 2296.29 | 2.68 | 61.97 | 6.09 | 6.31 | 6.04 | 6.15 | 4.36 | 0.13 | 4.68 | 5.07 | 4.44 | 4.73 | 6.06 | 6.22 | 5.93 | 6.07 |
| 691-1* | Grt-bearing mica schist | 2457.26 | 2.85 | 66.17 | 6.42 | 6.68 | 5.51 | 6.20 | 17.51 | NA | 2.45 | 3.69 | 2.59 | 2.91 | 6.41 | 6.54 | 5.49 | 6.15 |

*) Samples measured by Wenning et al. (2016), data recalculated.



## 14 Appendices

700 **Table A1.** List of investigated core samples, their exact core position, and associated International Geo Sampling Number (IGSN). The section tops refer to the meter-corrected depth from the operation data sets (Lorenz et al., 2019). FK – samples newly measured for this study; QW – samples originally measured by Wenning et al. (2016).

| Core | Section | Box | Slot | Section Top (m) | Sample Top (cm) | Sample Bottom (cm) | Examiner | IGSN |
|---|---|---|---|---|---|---|---|---|
| 106 | 1 | 102 | 1 | 402.80 | 20 | 40 | FK | BGRB5054RXG9401 |
| 143 | 1 | 135 | 1 | 501.76 | 20 | 40 | FK | BGRB5054RXI9401 |
| 149 | 4 | 141 | 4 | 522.22 | 9 | 29 | FK | ICDP5054EXK6601 |
| 193 | 2 | 184 | 2 | 649.51 | 0 | 15 | QW | ICDP5054EXL6601 |
| 243 | 2 | 233 | 2 | 790.49 | 65 | 80 | QW | ICDP5054EXN6601 |
| 361 | 2 | 343 | 3 | 1120.76 | 55 | 82.6 | FK | BGRB5054RXP9401 |
| 487 | 1 | 467 | 1 | 1490.97 | 62 | 87 | QW | ICDP5054EXS6601 |
| 556 | 2 | 528 | 3 | 1689.84 | 0 | 20 | QW | ICDP5054EXU6601 |
| 569 | 2 | 542 | 3 | 1743.89 | 27 | 42 | FK | ICDP5054EXV6601 |
| 593 | 4 | 575 | 4 | 1880.83 | 60 | 75 | FK | ICDP5054EX47601 |
| 631 | 1 | 627 | 5 | 2096.96 | 60 | 78 | QW | ICDP5054EXY6601 |
| 641 | 5 | 642 | 4 | 2160.49 | 0 | 20 | FK | BGRB5054RXY9401 |
| 651 | 5 | 656 | 1 | 2220.29 | 10 | 30 | FK | BGRB5054RX0A401 |
| 661 | 3 | 669 | 4 | 2278.43 | 77 | 95.8 | FK | BGRB5054RX1A401 |
| 664 | 2 | 673 | 4 | 2295.51 | 70 | 85 | FK | ICDP5054EX6A601 |
| 691 | 1 | 711 | 2 | 2456.64 | 54 | 69 | QW | ICDP5054EX27601 |