# Peer review of "Correlation of core and downhole seismic velocities in high-pressure metamorphic rocks: A case study for the COSC-1 borehole, Sweden"

_Solid Earth, 2019_

## Referee Comment (RC1) · Anett Blischke (Referee) · 16 Dec 2019

Interesting study that adds to the borehole / core in-situ lab experiments for metamorphic rock settings, which is important for extending the global calibration database for deeply buried metamorphic rocks and their understanding.

The study's aim was however (as firstly stated in the introduction) to improve the imaging of thrust zones and the understanding of the deeper orogenic processes and tectonic evolution? How does the manuscript relate to that project objective? I think this needs a small revision of the introduction and discussion sections to fit the study that in itself has a good closure.

[Figure]

Attached is a list of suggestions and comments that hopefully will be of use to make this a good paper and contribution.

Looking forward to see the revised manuscript.

Please also note the supplement to this comment:
https://www.solid-earth-discuss.net/se-2019-161/se-2019-161-RC1-supplement.pdf

[Figure]

**Supplement:**

Solid Earth: Manuscript Number: https://doi.org/10.5194/se-2019-161

**Correlation of core and downhole seismic velocities in high-pressure metamorphic rocks: A case study for the COSC-1 borehole, Sweden**
*Manuscript by Kästner et al. 2019*

**General remarks and main points:**

- Interesting study that add to the borehole / core in-situ lab experiments for metamorphic rock settings, which is important for extending the global calibration database for deeply buried metamorphic rocks and their understanding.

- This aim was however (as firstly stated in the introduction) to improve the imaging of thrust zones and the understanding of the deeper orogenic processes and tectonic evolution? How does the manuscript relate to that project objective? I think this needs a small revision of the introduction to fit the study that in itself has a good closure.

- I bring in quite a few suggestions, but hope they help to make this a good paper and contribution.

- Looking forward to see the revised manuscript.

**Abstract:**
Has all the info, still possibly sort the sentences.
- Where?
- What objective?
- Doing What?
- How?
- Resulting?
- What didn´t work?
- What did?
- Why are the study results important?

Please re-arrange specifically this passage, I am getting confused what are the results that didn´t worked and what did. Always better to end on the results that did work:
"*The core and downhole velocities deviate by up to 2 km/s. However, velocities of mafic rocks are 15 generally in close agreement. Seismic anisotropy increases from about 5 to 26 % at depth, indicating a transition from gneissic to schistose foliation. We suggest that differences in the core and downhole velocities are most likely the result of microcracks mainly due to depressurization. Thus, seismic velocity can help to identify mafic rocks on different scales whereas the velocity signature of other lithologies is obscured in core-derived velocities. Metamorphic foliation on the other hand has a clear expression in seismic anisotropy.*"

Please just refer to the *COSC-1 borehole* consistently.

**Introduction:**

General, please check that references are placed, were facts and introductions are stated.

Your primary objective is to " *… to improve our understanding of the deeper orogenic processes and tectonic evolution."* (*first paragraph*)

Than follows the geophysical experiments that led to this study (REF?), specifically seismic reflection data and imaging of that thrust zone (REF?), and how does better sub-surface imaging than improve the understanding of the tectonic evolution based on core data?

Suggest to re-phrase the primary objective that than follows well into the paragraphs (L33-39), as here it is a lithological / stratigraphic objective and not the thrust zone is described. Knowing the stratigraphy and the rocks petrophysical properties would lead to better seismic reflection data processing and imaging for example.

If there are only 2 primary projects of this kind KTB or the CCSD, why not say so and spell them out? What about the study by Zappone et al. (2000) for the Iberian, or the Kola Borehole Kern et al. (2001)?

*Kern, H. & Popp, Till & Gorbatsevich, Feliks & Zharikov, Andrey & Lobanov, K. & Smirnov, Yu. (2001). Pressure and temperature dependence of V P and V S in rocks from the superdeep well and from surface analogues at Kola and the nature of velocity anisotropy. Tectonophysics. 338. 113-134. 10.1016/S0040-1951(01)00128-7.*

L48-51: This sentence I would suggest moving up front to follow with the supporting role of this project to better understanding of thrust zone and metamorphic settings.

L53-58: Isn´t this better placed in the methods section?

Section 1.1: This is a good concise overview but using the geological map and cross-section with the COSC-1 borehole projected on it, would really help to get the borehole´s geological setting´s placed in the readers head, especially if one isn´t that familiar with the area. This gives an option to show that main subdivisions described by Lorenz et al. (2015a) that leads well into the smaller scale core-based experiment.

L75: What type of deformations (fracturing / folding / cataclastic)?

**Data and methods:**

Please be more specific in describing what downhole logging data in the intro, you have them in Table 1. It would be good to briefly just state that these included short-spaced sonic and zero-offset VSP in the text with the appropriate referencing and reference to Figure 2, this way it´s clear from the start of reading this chapter.

L98: see Table 2 and Appendix A1. I am a bit missing a Figure that shows the known geology with the core section and sample location. This is a preference for people that prefer to see the graphic setup of the borehole samples. Just a suggestion, but this way it would be easier to follow who measured what at sample depth, with higher and lower reflectivity zone based on VSP, etc.?. This could even be part of Appendix A1 if the number of figures as to stay. Please see Figures 2 by Zappone et al. (2000).

L104-107: Could you indicate this in Figure 4, there would be enough space for the Core MSCL image, indicating the xyz structural axis to the foliation plane. Please see Figures 2 by Zappone et al. (2000).

Good explanation of the Figure 3 and the method.

L131-133: What temperature was at the 2500m? $T_{2500m}$ = (20 °C/km * 2,5 km) + 6,4 °C = 56.4°C?  I am just wondering as Table 2 indicates a depth range 400-2460m, which in turn would indicate a general linear in-situ temperature range between 14,4-55,6 °C. So why is room temperature acceptable, or has it been shown that temperature does not affect the measurements. If so please state and reference that.

Based on lab work by Mobarek (1971), would the Vp values be slightly low based on temperature increase. Of course, those tests were done on dry sandstone, and it would be good at least to describe how temperature would affect the lab results.

Possibly use ranges from similar studies in comparison (e.g. Iberian Peninsula)?

Motra, Hem & Stutz, Hans. (2018). Geomechanical Rock Properties Using Pressure and Temperature Dependence of Elastic P- and S-Wave Velocities. Geotechnical and Geological Engineering. 10.1007/s10706-018-0569-9.

L144-154: So what are you saying, are you applying this method or not? It´s just the explanation and reasoning – the data trend measured vs. empirical looks convincing – perhaps just rephrase slightly to be clear.

L156: Different in what?

Section 2.2: As you do not reference the setup anywhere in this section, either add a figure explaining this as in Figure 3 for the lab setup or point to the appropriate reference that one can go to for understanding the setup. Is this your method, then state this, or refer to the method shown as a reference? You have done that for your lab setup and the VSP.

L179: … accordingly to what?

L214: … at 0,5 m spacing?

L219: I would leaved this and state "used in our study". If you start mentioning "best case" than this naturally follows the question, what the low and high cases are that need explaining.

**Results:**

L224: Here it would have been nice to see the data plotted of Table 2, as described as the applied method on Figure 5. These are the main results that all following conclusion is based on.

At the moment the Table 2 has only the final results Vp0, VpAP, VpLP vs. build up pressure for each vector and mean, based on your measurements and calculations.

Did you double check by including measurements for increased pressure that gives a step by step series measured Vp that would demonstrate with you data what was explained in Figure 5 and the methods?

Please see example here:
https://academic.oup.com/gji/article-pdf/187/3/1393/1694975/187-3-1393.pdf

L232: Do you mean "core plug axial measurements"?

L250-261: Here it would be good to point out that core-derived Vp realtes to the whole core log measurements at surface conditions, whereas the Sonic-VSP Log are measured has the hydrostatic pressure in the borehole and the insitu-rock.

I would suggest to point out the depth intervals, where the logged rock Velocity is opposite in the general trend of the VSP-Log data that follows the lithology changes - add density log alongside? What about fault / fracture zones that would stand out of the rock matrix investigated?

L262-263: Please be specific that the reader can follow ... e.g. VpAP with the core logged velocity at surface, the VpLP to the VSP-Log data.

I would suggest pointing out the samples that are outliers / slightly off, e.g.  106.1; 143.1; 243.2; 361.2; 641.5; 661.3; or 691.1
This is more specific than to negate an entire interval, as the shallow data do not miss-match that much in comparison to the deeper interval.
You are doing this for the VpAP in the next section below.

L269: Why might this be? Are those sample much fractured?

L290: What do you mean with improper relationship? The matches are close-reasonable for examples B, E,F and D, but samples A and C are consistent lower. In comparison the the lithology is that seen along the borehole at other depths as well?

L295: Why keep working with that example if you do not show it? Is it possible to add the example to the display in Figure 11?

L:303: Why might this be? Please explain.

L-329-330: Is that also in reference to your final figure 12? You are using 106-1 to 193-2 format for the other intervals with selected samples.

L346: Just Figure 10?
Isn´t this best displayed on Figure 12?

L365: What about saturated micro-fractures that are measured as well within the matrix rock?

L374: ... for X points out of X of VpLP.

L392: Are you talking about the Core Log" or Vp0, VpAP, and VpLP as a group of VpLP specifically?

Might be good to specify at the beginning of the paragraphe, so the reader doesn´t mix up the two data sets.

L418: Possibly marked these 10% as depth intervals on Figure 10 and 12.

L460: Definitely revise your introduciton to focus the study on the matrix primarily and influences of fractures / micro-fracters,

**Figures and table:**

- Figures and Tables are clearly structured, and features displayed well visible, still here are a few comments and suggestions.

- Figure 1:
    - The figure looks too much as a copy – past. You could ask to get a GIS version / emf draft of that map and leave out all the lines and info that doesn´t matter, such as roads, power lines. Just focus on the geological- and tectonic, and borehole location. Standard is – if you have something displayed on your map, you should include that in the legend.

    - Are any profile section available to show how the borehole is placed and intersects the thrust zone? There are quite a few references listed that show that this should be available (Gee et al., 1985a,b; 2008, 2010; or Hedin et al., 2014, 2016, etc.).

      You state in the abstract "*Previous seismic investigations of the Seve Nappe Complex have shown indications for a strong but discontinuous reflectivity of this thrust zone, which is only poorly understood.*" Seeing this, as the reader I would expect a section / profile that shows that for the introduction.

      It´s just nicer to know really where the borehole is located and the general stratification that has been worked out already (Lorenz et al., 2015a,b, 2019; Krauss et al., 2017; Wenning et al, 2017; etc.)

    - Legend text, would be good to use emf-format; include reference to the map as a publication. The geological survey maps do in general have a publication reference.

- Figure 2:

- o Please add referencing for the Downhole-logging data input.

- o Could you indicate this in Figure 4, there would be enough space for the Core MSCL image, indicating the xyz structural axis to the foliation plane. Please see Figures 2 by Zappone et al. (2000).

- Figure 4:
  - o Please add reference for density log data source.

- Figure 5:
  - o Please add the reference that the method used is after Ji et al. (2007).

- Figures 10 & 11:
  - o Please increase text size in Figures 10 and 11 similar to Figure 12

- Figure 12:
  - o Please add the VpLP results to the plot.

- If it is not too much trouble, please add the legends on each figure, there is enough space.

- Table 1:
  - o Please add references to all downhole logging data that were not part of this study but used. This helps to keep this separate, what is new data and what is part of this study.

*A few first Questions that came to mind:*

- How does the metamorphic facies Vp results compare to other similar projects (e.g. good comparison studies doi:10.1029/2006JB004867, 2007 or doi:10.1144/GSL.SP.1998.136.01.9, 1998)

- Uncertainty analysis is listed in references, but how was it implemented?

- As a connection to structural changes of the rock is mentioned as the primary reason for anisotropy, why not mark main structural intervals on Figure 10, if fractures have been analysed?

- What is similar / different to the Chinese CCSD borehole experiment or the southern Iberian Peninsula?

  If dissimilar, are local settings governing the results? Just to mind the statement that this method would be a good tool for similar cases.

---

## Referee Comment (RC2) · Anonymous Referee #2 · 2 Jan 2020

Following the developments in resources exploration industry, scientific studies using core-log-seismic data integration were progressed gradually over three decades under the scientific drilling programs as standard use on almost every projects. However, quality of the data and integration process were limited on use of tools, operational difficulties, particularly in ocean drilling projects and for hard rocks. Hence, there are very few studies on hard rock data integration and this research is one of the few case using complete set of core, log and seismic data in high quality data as well as supported further experiment for integration.

Please also note the supplement to this comment:

[Figure]

https://www.solid-earth-discuss.net/se-2019-161/se-2019-161-RC2-supplement.pdf

[Figure]

**Supplement:**

[revised manuscript text omitted]

---

## Author Comment (AC1) · 14 Feb 2020

Following the developments in resources exploration industry, scientific studies using core-log-seismic data integration were progressed gradually over three decades under the scientific drilling programs as standard use on almost every projects. However, quality of the data and integration process were limited on use of tools, operational difficulties, particularly in ocean drilling projects and for hard rocks. Hence, there are very few studies on hard rock data integration and this research is one of the few case using complete set of core, log and seismic data in high quality data as well as supported further experiment for integration.

**Author's response**

We thank the reviewer for reviewing our manuscript submitted for publication in Solid Earth. The reviewer suggested a few minor modifications of the manuscript. According to the suggested amendments, we list the reviewer comments (RC2) and our proposed changes to the manuscript (AR) in the following:

**L20:**
*RC2: Core-log-integration using synthetic seismograms requires wireline logging data and mafic lithologies.*
AR: This is correct. This sentence is perhaps not clearly formulated. We removed it.

**L167:**
RC2: Rephrase to "[...] accuracy of the pulse arrival picks [...]"
AR: Amended.

**L208:**
RC2: Remove "equals"
AR: Done.

**L244:**
RC2: Amend sentence to "[...] low values, agreeing well with the velocities measured on core [...]"
AR: Amended.

**L257:**
RC2: Insert missing comma.
AR: Done.

**L289:**
RC2: Insert "with"
AR: Done.

**L290:**
RC2: Replace "the" with "these"
AR: Amended.

**L328:**
RC2: Rocks of the present study?
AR: Here, we are referring to the rocks found in the COSC-1 lithology. However, of course, this in particular is only true for the analyzed samples. We rephrased this for clarification: "This suggests a strong structural dependence of the seismic velocities for the rocks of the Seve Nappe drilled by the COSC-1 borehole."

**L335:**
RC2: "Here"?
AR: We rephrased this line for clarification: "The presented anisotropy-depth profile […]". We also added a reference to the referred figure.

**L336:**
RC2: Remove "Nevertheless"
AR: Amended sentence to: "Despite of large data gaps, […]"

**L372:**
RC2: Respell to "mantle-driven"
AR: Done.

**L390:**
RC2: Slightly low values or low values?
AR: Because of the insufficient closure microcracks we infer slightly lower velocities for the felsic gneisses than expected. We rephrased the sentence for clarification: "[…] result in lower values for […]"

**L444:**
RC2: "anisotrophies"
AR: We prefer to stay using the singular and respelled it accordingly to "anisotropy".

---

## Author Comment (AC2) · 14 Feb 2020

**Author response to "Correlation of core and downhole seismic velocities in high-pressure metamorphic rocks: A case study for the COSC-1 borehole, Sweden"**

Manuscript by Kästner et al. 2019

**Referee: Anett Blischke (ÍSOR Iceland Geosurvey)**

**General remarks and main points:**

- Interesting study that add to the borehole / core in-situ lab experiments for metamorphic rock settings, which is important for extending the global calibration database for deeply buried metamorphic rocks and their understanding.
- This aim was however (as first stated in the introduction) to improve the imaging of thrust zones and the understanding of the deeper orogenic processes and tectonic evolution? How does the manuscript relate to that project objective? I think this needs a small revision of the introduction to fit the study that in itself has a good closure.
- I bring in quite a few suggestions, but hope they help to make this a good paper and contribution.
- Looking forward to see the revised manuscript.

We thank the reviewer for the comprehensive review of our submitted manuscript, giving a number of comments and suggestions for improvements. In the following, we attempt to respond to each of the comments and amendments, and, where applicable, we propose our changes for the revised manuscript. We feel that the presented changes will greatly improve our submission in order to provide it in a format acceptable for publication within this Journal.

**Abstract:**

Has all the info, still possibly sort the sentences.

- Where?
- What objective?
- Doing What?
- How?
- Resulting?
- What didn't work?
- What did?
- Why are the study results important?

Please re-arrange specifically this passage, I am getting confused what are the results that didn't work and what did. Always better to end on the results that did work:

"The core and downhole velocities deviate by up to 2 km/s. However, velocities of mafic rocks are generally in close agreement. Seismic anisotropy increases from about 5 to 26 % at depth, indicating a transition from gneissic to schistose foliation. We suggest that differences in the core and downhole velocities are most likely the result of microcracks mainly due to depressurization. Thus, seismic velocity can help to identify mafic rocks on different scales whereas the velocity signature of other lithologies is obscured in core-derived velocities. Metamorphic foliation on the other hand has a clear expression in seismic anisotropy."

To avoid any confusion, we have re-arranged the abstract paragraph accordingly, as proposed by the reviewer:

"[...] For some intervals of the COSC-1 borehole, the core and downhole velocities deviate by up to 2 km/s. These differences in the core and downhole velocities are most likely the result of microcracks mainly due to depressurization. However, the core and downhole velocities of the intervals with mafic rocks are generally in close agreement. Seismic anisotropy measured on laboratory samples increases from about 5 to 26 % at depth, correlating with a transition from gneissic to schistose foliation. Metamorphic foliation on the other hand has a clear expression in seismic anisotropy. These results will aid in the evaluation of core-derived seismic properties of high-grade metamorphic rocks at the COSC-1 borehole and elsewhere."

Please just refer to the COSC-1 borehole consistently.

We have revised and adjusted all references to the borehole to "COSC-1".

In line 47 we change "COSC project" to "COSC-1 drilling project".

**Introduction:**

General, please check that references are placed, were facts and introductions are stated.

Your primary objective is to "... to improve our understanding of the deeper orogenic processes and tectonic evolution." (first paragraph)

Then follows the geophysical experiments that led to this study (REF?), specifically seismic reflection data and imaging of that thrust zone (REF?), and how does better sub-surface imaging than improve the understanding of the tectonic evolution based on core data?

**See comment below.**

Suggest to rephrase the primary objective that than follows well into the paragraphs (L33-39), as here it is a lithological / stratigraphic objective and not the thrust zone is described. Knowing the stratigraphy and the rocks petrophysical properties would lead to better seismic reflection data processing and imaging for example.

We agree that the objective of this particular study is not clearly separated from the primary ("long-term") objective of the COSC drilling project. As pointed out by the reviewer, the ultimate aim addressed by the COSC drilling project is the understanding of the deeper orogenic processes and tectonic evolution of the Scandinavian Caledonides. Seismic reflection data provides one tool to image subsurface and to interpret these structures (including thrust zones and nappe stacks) at depth. Especially, by knowing the physical properties of the associated rocks at depth this can help to better constrain reflections and aid the processing and interpretation of these reflection profiles.

We have modified this part of the introduction in order to clearly point out which is the long-term project aim and how our study contributes to it (objective of this study versus COSC drilling project aims). Essentially, this has required some sentences to be rephrased or moved.

If there are only 2 primary projects of this kind KTB or the CCSD, why not say so and spell them out? What about the study by Zappone et al. (2000) for the Iberian, or the Kola Borehole Kern et al. (2001)? (Kern, H. & Popp, Till & Gorbatsevich, Feliks & Zharikov, Andrey & Lobanov, K. & Smirnov, Yu. (2001). Pressure and temperature dependence of V P and V S in rocks from the superdeep well and from surface analogues at Kola and the nature of velocity anisotropy. Tectonophysics. 338. 113-134. 10.1016/S0040-1951(01)00128-7.)

Certainly, there are more than two studies of such kind. Our purpose was to highlight especially these two, which are of comparable geological/tectonic setting with a similar high core recovery.

We agree, however, we have added additional references in order to provide a more concise overview of the related literature and background.

We therefore have explicitly named these projects and, furthermore, added the following references:

- Golovataya, O. S., Gorbatsevich, F. F., Kern, H. and Popp, T.: Properties of some rocks from the section of the Kola ultradeep borehole as a function of the P-T parameters, Izv. Phys. Solid Earth, 42(11), 865–876, doi:10.1134/S1069351306110012, 2006.
- Sun, S., Ji, S., Wang, Q., Xu, Z., Salisbury, M. and Long, C.: Seismic velocities and anisotropy of core samples from the Chinese Continental Scientific Drilling borehole in the Sulu UHP terrane, eastern China, J. Geophys. Res. Solid Earth, 117(B1), n/a-n/a, doi:10.1029/2011JB008672, 2012.
- Kern, H., Schmidt, R., Drilling, T. P.-S. and 1991, U.: The velocity and density structure of the 4000 m crustal segment at the KTB drilling site and their relationship to lithological and microstructural, Sci. Drill., 2, 130–145, 1991.
- Elbra, T., Karlqvist, R., Lassila, I., Haeggström, E. and Pesonen, L. J.: Laboratory measurements of the seismic velocities and other petrophysical properties of the Outokumpu deep drill core samples, eastern Finland, Geophys. J. Int., 184(1), 405–415, 2011.

**L48-51:** This sentence I would suggest moving up front to follow with the supporting role of this project to better understanding of thrust zone and metamorphic settings.

Moved sentence up to line 35.

**L53-58:** Isn't this better placed in the methods section?**

In order to give the reader short agenda and an idea of the following sections, we have briefly introduced the applied methods. We think this is a worthwhile extension of the introduction.

**Section 1.1:** This is a good concise overview but using the geological map and cross-section with the COSC-1 borehole projected on it, would really help to get the borehole's geological settings placed in the reader's head, especially if one isn't that familiar with the area. This gives an option to show that main subdivisions described by Lorenz et al. (2015a) that leads well into the smaller scale core-based experiment.

We agree and have added an additional subfigure to get the reader more familiar with the regional setting. See also comment in the figures subsection further below.

**L75:** What type of deformations (fracturing / folding / cataclastic)?

Rocks of the COSC-1 borehole predominantly exhibit ductile shear deformation with signs of mylonitic deformation and micro-folding.

Amended sentence: "[...] (2) an extensive (ductile shear) deformation zone prevails between [...]"

**Data and methods:**

Please be more specific in describing what downhole logging data in the intro, you have them in Table 1. It would be good to briefly just state that these included short-spaced sonic and zero-offset VSP in the text with the appropriate referencing and reference to Figure 2, this way it's clear from the start of reading this chapter.

We have rephrased the sentence to: "[...] from a multi-sensor core log (MSCL) and downhole data from the COSC-1 borehole including short-spacing sonic log and zero-offset VSP".

**L98:** see Table 2 and Appendix A1. I am a bit missing a Figure that shows the known geology with the core section and sample location. This is a preference for people that prefer to see the graphic setup of the borehole samples. Just a suggestion, but this way it would be easier to follow who measured what at sample depth, with higher and lower reflectivity zone based on VSP, etc.?. This could even be part of Appendix A1 if the number of figures has to stay. Please see Figures 2 by Zappone et al. (2000).

We understand the reviewers suggested to show the geology of the samples in a figure. We have referred to Figure 10, where the lithological units, and the depth of the samples is shown.

**L104-107:** Could you indicate this in Figure 4, there would be enough space for the Core MSCL image, indicating the xyz structural axis to the foliation plane. Please see Figure 4 by Zappone et al. (2000).

We agree that it is useful for the reader to understand the relation between the structural axes and the core plug measurements. Indicating this in Figure 2, however, next to the Core MSCL picture might give a wrong impression that the core/borehole is always perpendicular to the foliation plane, which is not always the case. Especially, with Figure 2 we intend to provide the more general case highlighting only the different scale and measurement conditions. Thus, we prefer to rather implement the structural coordinate system in another figure together with a raw data example (see comment on Figure 5 and L224).

Good explanation of the Figure 3 and the method.

**L131-133:** What temperature was at the 2500m?  $T_{2500m} = (20 \text{ °C/km} * 2,5 \text{ km}) + 6,4 \text{ °C} = 56.4 \text{ °C}? I am just wondering as Table 2 indicates a depth range 400-2460m, which in turn would indicate a general linear in-situ temperature range between 14,4-55,6 °C. So why is room temperature acceptable, or has it been shown that temperature does not affect the measurements. If so please state and reference that.$

For this study, of highly consolidated metamorphic rocks, thermal effects on velocity and anisotropy can be neglected (e.g., Kern 1990). Especially, for the investigated depth and temperature range these effects are small compared to pressure effects (compare with Zappone et al. (2000)). The following paragraph clarifies this.

We have added the following paragraph, for clarification:

"Generally, seismic velocities decrease with temperature (e.g., Schön (1996), Motra and Stutz (2018)). However, at very low temperatures (

**Figure 1:** Overview of the regional setting and study area. (a) Tectonostratigraphic division of the central Scandinavian Caledonides (Gee et al. (2010); Lorenz et al., 2015); (b) Bedrock map with location of the COSC-1 borehole (colors modified; SGU Map Service; Strömberg et al., 1994); (c) Seismic cross-section indicated in (b), showing a part of the COSC seismic profile (Hedin et al., 2012) with the COSC-1 borehole penetrating the highly reflective Lower Seve Nappe (adapted from Juhlin et al., 2016).

 Is there any profile section available to show how the borehole is placed and intersects the thrust zone? There are quite a few references listed that show that this should be available (Gee et al., 1985a,b; 2008, 2010; or Hedin et al., 2014, 2016, etc.).

Yes, there is. We have included it as described above.

You state in the abstract "Previous seismic investigations of the Seve Nappe Complex have shown indications for a strong but discontinuous reflectivity of this thrust zone, which is only poorly understood." Seeing this, as the reader I would expect a section / profile that shows that for the introduction.

It's just nicer to know really where the borehole is located and the general stratification that has been worked out already (Lorenz et al., 2015a,b, 2019; Krauss et al., 2017; Wenning et al, 2017; etc.)

Please, see comment above.

• Legend text, would be good to use emf-format; include reference to the map as a publication. The geological survey maps do in general have a publication reference.

We were not able to find any explicit map reference on the SGU webpage. However, the map is likely based on several mappings in the area over several years by different people (Lorenz, personal comm.). With respect to Gee et al. (2010, Fig. 8) the map is most likely based on the geological map from Strömberg et al. (1994). We have therefore added Strömberg et al. (1994) to the figure references. Please, see also general comment to Figure 1, further above.

- Figure 2:
  - Please add referencing for the Downhole-logging data input.

Figure 2 is a generic explanation just to show the integration of the three methods (core, downhole logging and VSP) related to a specific physical parameter used in this study, namely the compressional wave velocity (Vp). We used velocity, because it was the common physical parameter for all three methods. This same method can be applied using other types of physical properties (e.g., density). So we prefer to not add additional references.

 Could you indicate this in Figure 4, there would be enough space for the Core MSCL image, indicating the xyz structural axis to the foliation plane. Please see Figures 2 by Zappone et al. (2000).

We have indicated the core plug orientations and structural coordinates in Figure 2 as suggested. The figure capture text has been adjusted accordingly as shown below. Please also refer to comment on L104.